# Geometry-aware Bayesian Optimization in Robotics using Riemannian Matérn Kernels

**Noémie Jaquier**[*1]    **Viacheslav Borovitskiy**[*2, 6]    **Andrei Smolensky**[2]
**Alexander Terenin**[3, 4]    **Tamim Asfour**[1]    **Leonel Rozo**[5]
[1]Karlsruhe Institute of Technology    [2]St. Petersburg State University
[3]University of Cambridge  [4]Imperial College London  [5]Bosch Center for Artificial Intelligence
[6]St. Petersburg Department of Steklov Mathematical Institute of Russian Academy of Sciences

**Abstract:** Bayesian optimization is a data-efficient technique which can be used for control parameter tuning, parametric policy adaptation, and structure design in robotics. Many of these problems require optimization of functions defined on non-Euclidean domains like spheres, rotation groups, or spaces of positive-definite matrices. To do so, one must place a Gaussian process prior, or equivalently define a kernel, on the space of interest. Effective kernels typically reflect the geometry of the spaces they are defined on, but designing them is generally non-trivial. Recent work on the Riemannian Matérn kernels, based on stochastic partial differential equations and spectral theory of the Laplace–Beltrami operator, offers promising avenues towards constructing such geometry-aware kernels. In this paper, we study techniques for implementing these kernels on manifolds of interest in robotics, demonstrate their performance on a set of artificial benchmark functions, and illustrate geometry-aware Bayesian optimization for a variety of robotic applications, covering orientation control, manipulability optimization, and motion planning, while showing its improved performance.

**Keywords:** Bayesian optimization, Matérn kernels, Riemannian manifolds

## 1  Introduction

Fast and data-efficient adaptation is a key challenge in a large range of robotics applications, owing to the need to handle sensor noise, model uncertainty, and generalize to unforeseen settings. In this context, Bayesian optimization [46] is a technique of growing interest in robotics due to its ability to efficiently solve challenging optimization problems, including controller tuning [5, 33], policy adaptation [4, 15], and robot design [42]. This makes it an attractive tool for settings where data-efficiency is of key interest, and large-scale methods such as deep reinforcement learning are not presently viable. Bayesian optimization algorithms are generally powered by probabilistic models based on Gaussian processes that rely on inductive biases encoded by kernels. Improving performance and scalability of such models via a number of different strategies has therefore been a central goal pursued by the Bayesian optimization and Gaussian process literatures.

Inductive bias in robotics can be introduced by exploiting domain knowledge about the task at hand. For example, task-specific kernels were proposed by Antonova et al. [5] by learning a distance metric via simulated bipedal locomotion patterns to optimize gait controllers, and by Rai et al. [38] using gait feature transformations to design kernels to optimize locomotion controllers. More generic kernels were introduced by Marco et al. [33], Marco et al. [34], and Wilson et al. [53] for policy optimization by incorporating linear-quadratic control structures and trajectory information into the kernel design. While these kernels can be exploited for robot control or policy search with many different tasks and systems, their use is specific to the problems they were designed for.

A more general approach is to carefully select the *domain* of the function being optimized, thus incorporating information about the geometry of the search space into the optimization algorithm.

---

[*]Equal contribution. Correspondence to: noemie.jaquier@kit.edu, viacheslav.borovitskiy@gmail.com.
Code at https://github.com/NoemieJaquier/MaternGaBO and video at https://youtu.be/6awfFRqP7wA.

5th Conference on Robot Learning (CoRL 2021), London, UK.

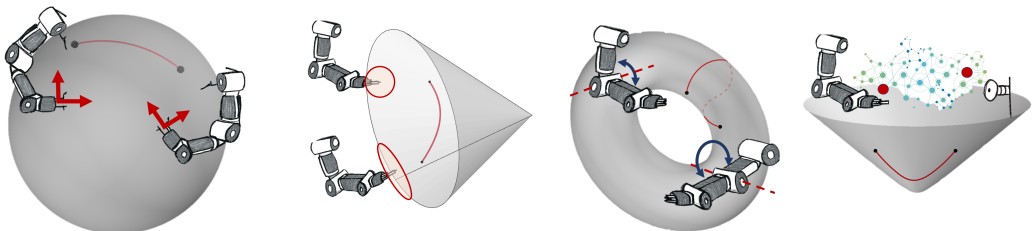

Figure 1: Illustration of different Riemannian manifolds of interest in robotics: a unit sphere $\mathbb{S}^d$, symmetric positive definite matrices $\mathcal{S}_{++}^d$, a torus $\mathbb{T}^d$, and the Lorentz model of hyperbolic geometry in $\mathcal{H}^2$ (from left to right). Red solid lines depict geodesics between two (black) points on the manifold.

Many quantities of interest in robotics carry such geometric information: three-dimensional rotations can be viewed as elements of the Lie group $\mathrm{SO}(3)$ [20] or the sphere $\mathbb{S}^3$ [51], control gains, inertia and manipulability ellipsoids lie in the manifold of symmetric-positive-definite matrices $\mathcal{S}_{++}^d$, while the joint configuration of a $d$-degree-of-freedom robot with revolute joints may be viewed as a point on a torus $\mathbb{T}^d$, i.e. the product of $d$ circles [28]: see Figure 1 for illustrations. Incorporating this geometric structure into Bayesian optimization has recently been shown to improve its performance on a number of tasks [27], despite the use of a naïve geometrical approximation in the kernel design.

Geometry-aware Bayesian optimization is therefore a promising emerging tool for general use in robotics. One of the main difficulties in developing this tool is that defining geometry-aware Gaussian process models is technically non-trivial: classic Euclidean methods ignore the parameter space geometry, and naïve geometric approximations do not capture the manifold's geometric structure. Fortunately, mathematical theory and techniques for building the necessary kernels are beginning to emerge [8, 30]. In particular, the work of Borovitskiy et al. [8] on Riemannian Matérn kernels provides mathematically-sound tools for calculating geometry-aware kernels by building on stochastic partial differential equations and spectral theory of the Laplace–Beltrami operator. This provides the key ingredients to deploy geometry-aware Bayesian optimization on practical robotics problems.

Building on the performance improvements in Jaquier et al. [27] and the theoretically-grounded Matérn kernels proposed in Borovitskiy et al. [8], we present a more general kernel formulation for a wide range of Riemannian manifolds of interest in robotics. Our approach uses the fact that a Matérn kernel can be formulated as the integral of a squared-exponential kernel: this idea enables us to define Riemannian Matérn kernel on non-compact manifolds, generalizing the work of Borovitskiy et al. [8] which applies only to compact manifolds. We introduce new Riemanian Matérn kernels for the special orthogonal group, manifold of symmetric positive definite matrices, and hyperbolic space, and study their use in robotic applications. This more general formulation avoids the need to determine valid length scales as in prior work [27], provides theoretical guarantees of positive-definiteness, and makes kernel smoothness optimization possible. We benchmark these kernels on a set of artificial functions and on a number of robotics problems of interest to practitioners.

## 2 Geometry-aware Bayesian Optimization

Bayesian optimization [45] is a class of sequential search algorithms for finding a global minimizer $\boldsymbol{x}^* = \arg\min_{\boldsymbol{x} \in \mathcal{X}} f(\boldsymbol{x})$ of an unknown objective function $f$ defined over a domain of interest $\mathcal{X}$. The function $f$ is not observed directly: instead, at each step the algorithm must select a query point $\boldsymbol{x} \in \mathcal{X}$ at which to evaluate $f$. Moreover, it is usually assumed that the algorithm only observes a noisy value $y \in \mathbb{R}$ whose expectation $\mathbb{E}(y) = f(\boldsymbol{x})$ is the true function value at the chosen point. Based on the full set of observed evaluations, the algorithm must select the next query point $\boldsymbol{x}$.

The vast majority of Bayesian optimization algorithms employ Gaussian process models, which represent the known information about the objective function using a Gaussian process prior $f \sim \mathrm{GP}(\mu, k)$, with mean function $\mu : \mathcal{X} \to \mathbb{R}$ and symmetric-positive-definite kernel $k : \mathcal{X} \times \mathcal{X} \to \mathbb{R}$. The Gaussian process is updated by conditioning on observed data $(\boldsymbol{x}_i, y_i)_{i=1}^n$ using Bayes' Rule. The next query point is selected by optimizing a carefully-constructed acquisition function defined with respect to the resulting posterior distribution.

In robotics, the domain of interest $\mathcal{X}$ often possesses important non-Euclidean geometric structure. For example, when tuning robot controllers, the variables of interest include end-effector poses

lying on $\mathbb{R}^3 \times \mathbb{S}^3$ where $\mathbb{S}^d$ denotes the $d$-dimensional unit sphere, robot joint postures defined on a torus $\mathbb{T}^d = \mathbb{S}^1 \times \ldots \times \mathbb{S}^1$, as well as gain matrices and manipulability ellipsoids belonging to the space of symmetric-positive-definite matrices $\mathcal{S}_{++}^d$. In addition, when adapting parametric robot policies, some of their parameters such as mean vectors or covariance matrices lie in Riemannian manifolds. This inherent geometry is often overlooked in robotic applications [33, 40, 41]. In this paper, we demonstrate that incorporating the correct geometric structure of $\mathcal{X}$ into the Gaussian process is an important potential avenue for improving performance of Bayesian optimization.

Performing geometry-aware Bayesian optimization on Riemannian manifolds requires one to address two main challenges. Firstly, defining a valid Gaussian process prior is more difficult than in the Euclidean case. Secondly, numerical techniques such as training and optimizing the acquisition functions may require special considerations. We address these issues in the remainder of this section.

## 2.1 Gaussian Processes on Riemannian Manifolds

The key challenge in constructing Gaussian processes on non-Euclidean domains is in defining a valid kernel: seemingly obvious and innocuous-looking candidate expressions such as the squared exponential geodesic distance $\exp(-d_g(\boldsymbol{x}, \boldsymbol{x}')^2/\kappa)$ are not valid kernels for all length scales $\kappa > 0$ simultaneously [14]. In spite of this, they have been shown to be beneficial for practical robotics applications [27]: this motivates one to develop technically sound analogues. To tackle these issues, recent works on non-Euclidean Gaussian processes, including Lindgren et al. [30] and Borovitskiy et al. [7, 8], have developed provably well-posed techniques based on stochastic partial differential equations and spectral theory of the Laplace–Beltrami operator. We focus on the latter work as it does not require one to solve stochastic partial differential equations numerically.

The general theory of Matérn Gaussian processes on compact Riemannian manifolds from Borovitskiy et al. [8] introduces expressions for the kernels of these processes in terms of the Laplace–Beltrami eigenpairs of their respective manifolds. For more general settings including non-compact manifolds, such as the manifold of positive definite matrices and hyperbolic space, kernels can often be built by exploiting the connection between Riemannian squared exponential and heat kernels as fundamental solutions of heat equations. These can be connected with Matérn kernels by viewing them as certain integrals of squared exponential kernels [50]. We use this connection to introduce new Riemannian kernels on non-compact manifolds, and to further explore existing Riemannian kernels in Section 3.

## 2.2 Optimization of acquisition functions on manifolds

To use these Gaussian processes in a Bayesian optimization system, one must further be able to determine where to evaluate the unknown function and collect additional data. To do so, Bayesian optimization minimizes an acquisition function, which resolves the *explore-exploit tradeoff* [45] inherent in the problem due to lack of complete information about $f$. This is done, for instance, by prioritizing points which are known to perform well due to having high posterior mean, while also considering points whose performance is unknown due to high posterior variance. In this way, the probabilistic uncertainty given by the posterior Gaussian process is used to resolve the tradeoff.

In the geometric setting, Bayesian optimization therefore requires minimization of real-valued functions on manifold domains as an intermediate step. To do so, we leverage optimization algorithms on Riemannian manifolds [1] to maximize the acquisition function in a geometry-aware manner. These algorithms work with the Euclidean tangent space $\mathcal{T}_{\boldsymbol{x}}\mathcal{X}$ linked to each point $\boldsymbol{x}$ on the manifold $\mathcal{X}$, utilize gradients intrinsically defined as elements of the tangent space, and project the results of their computations back onto the manifold with the exponential map $\mathrm{Exp}_{\boldsymbol{x}} : \mathcal{T}_{\boldsymbol{x}}\mathcal{X} \to \mathcal{X}$. These different operations are determined based on the Riemannian metric with which the manifold is endowed.

In this paper, we employ trust region methods on Riemannian manifolds, as introduced by Absil et al. [2], to optimize the acquisition function at each iteration of Bayesian optimization. The recursive process used in trust region methods on Riemannian manifolds involves the same steps as its Euclidean counterpart, namely: (i) optimization of a quadratic subproblem $m_k$ trusted locally in a region around the iterate, (ii) updating of the trust region parameters, typically the trust-region radius $\Delta_k$, (iii) updating the iterates, where a candidate is accepted or rejected as function of the quality of the model $m_k$. For the cases where the space of interest is restricted to a subset of a Riemannian manifold, we employ a constrained trust region algorithm proposed by Jaquier and Rozo [25] that handles linear constraints. This recipe applies to almost all acquisition functions, including stochastic

ones via pathwise conditioning [54, 55], and allows us to introduce physical or safety constraints that naturally arise when interacting with physical systems and are crucial for robotics applications.

## 3  Matérn Kernels on Riemannian Manifolds

Training Gaussian processes, a crucial component of Bayesian optimization, requires us to be able to compute the kernel point-wise. This kernel encapsulates similarity information in the domain of interest: for domains with geometric structure, such as Riemannian manifolds, the kernel should reflect this structure in order to perform effectively. We now introduce geometric kernels that capture the structure of a number of spaces of interest in robotics, and study techniques for computing them.

As aforementioned, we focus on Riemannian Matérn kernels. These are based on the characterization of the widely used Euclidean Matérn Gaussian processes as solutions of a class of stochastic partial differential equations given by Whittle [52]. These equations naturally generalize to the Riemannian setting, and yield kernels which are well-defined for all Riemannian manifolds and hyperparameter values, unlike prior work [27] where restricting length scale range is required to guarantee positive definiteness. The main difficulty of this approach is that this definition is implicit and does not give practical expressions for computing the kernel without further considerations. For compact Riemannian manifolds, Borovitskiy et al. [8] prove expressions for Matérn kernels of the form

$$k_{\nu,\kappa,\sigma^2}(\boldsymbol{x},\boldsymbol{x}') = \frac{\sigma^2}{C_\nu} \sum_{n=0}^{\infty} \Phi_\nu(\lambda_n) f_n(\boldsymbol{x}) f_n(\boldsymbol{x}') \quad \Phi_\nu(\lambda_n) = \begin{cases} \left(\frac{2\nu}{\kappa^2} + \lambda_n\right)^{-\nu-\frac{d}{2}} & \nu < \infty, \\ e^{-\frac{\kappa^2}{2}\lambda_n} & \nu = \infty, \end{cases} \quad (1)$$

where $(\lambda_n, f_n)$ are Laplace–Beltrami eigenpairs ($\lambda_n \geq 0$), $\kappa$ is the length scale, $\sigma^2$ is the variance, $\nu$ is the smoothness and the constant $C_\nu$ is chosen so that the resulting kernels have average variance across the domain equal to $\sigma^2$. Note that despite notation $C_\nu$ depends on both $\nu$ and $\kappa$. The limiting kernel as $\nu \to \infty$, which we denote by $k_{\infty,\kappa,\sigma^2}$, can be viewed as the Riemannian analog of the Euclidean squared exponential kernel.

In practice, the kernel (1) and its squared exponential limit are computed by calculating the eigenpairs either analytically or numerically, and then truncating the infinite sum. Numerical computation of the eigenpairs amounts to approximating the manifold with a mesh and solving the eigenproblem using finite element methods [31]. Since this approach scales exponentially with manifold dimension, we do not focus on it in this work. Fortunately, owing to their extensive use in physics and other areas, mathematical techniques for evaluating the eigenpairs and simplifying the related expressions are available for many manifolds of interest in robotics: examples of this can be seen in the sequel. Although we only present the formulas for squared exponential (heat) kernels for the sake brevity, respective formulas for Matérn kernels hold as well.

The expression (1) does not apply for non-compact manifolds, although the implicit stochastic partial differential equation definition it is derived from is still valid. In this case, we leverage representations of Matérn kernels as integrals of squared exponential kernels, also known as heat kernels in the mathematical literature. In the Euclidean case, observe that

$$k_{\nu,\kappa,\sigma^2}(\boldsymbol{x},\boldsymbol{x}') = C \int_0^\infty u^{\nu-1} e^{-\frac{2\nu}{\kappa^2}u} k_{\infty,\sqrt{2u},\sigma^2}(\boldsymbol{x},\boldsymbol{x}') \, \mathrm{d}u, \quad (2)$$

where $C$ is given in (20). This can be obtained by writing the spectral measure of the Matérn kernel, the T distribution, as a gamma mixture of Gaussians. A similar relationship also holds in the compact Riemannian case: see Appendix B for details. More generally, it can be considered as the *definition* of a Matérn kernel in any setting where a suitable notion of a squared exponential or heat kernel is available (in this case we usually use $C = 1$), which is particularly useful in non-compact settings. Moreover, as shown in Appendix B, this kernel is positive definite whenever the respective heat kernel is positive definite. Therefore, it suffices to obtain expressions for the squared exponential kernel: analogous expressions for the Matérn kernel follow by integrating over length scales via (2). We thus focus on the heat kernel henceforth.[1]

These general expressions for Riemannian Matérn kernels allow us to build theoretically-sound kernels for the torus and the sphere. More importantly, we can introduce Riemannian formulations

---

[1]Note that it is usually beneficial to exploit explicit expressions for the Matérn kernels when they are available, e.g., (1), to alleviate additional numerical errors introduced by approximately computing the integral (2).

for kernels on non-compact manifolds such as the manifold of symmetric positive definite matrices, and $\mathcal{H}^d$. We now turn our attention to manifolds of key importance in robotics applications.

**Torus.** The torus $\mathbb{T}^d$ is defined as a product of circles $\mathbb{T}^d = \mathbb{S}^1 \times \ldots \times \mathbb{S}^1$, which naturally arises as, for example, the space of joint configuration of a $d$-degree-of-freedom robot. Functions on a torus $f : \mathbb{T}^d \to \mathbb{R}$ can be viewed as 1-periodic functions $f : \mathbb{R}^d \to \mathbb{R}$. Using this identification, the Laplace–Beltrami eigenfunctions on $\mathbb{T}^d$ are exactly those eigenfunctions of the Euclidean Laplacian which are 1-periodic: sines and cosines whose frequency is an integer multiple of $2\pi$. Substituting this into the general formula yields

$$k_{\infty,\kappa,\sigma^2}(\boldsymbol{x}, \boldsymbol{x}') = \frac{\sigma^2}{C_\infty} \sum_{\boldsymbol{\tau} \in \mathbb{Z}^d} e^{-2\kappa^2\pi^2\|\boldsymbol{\tau}\|^2} \cos(2\pi\langle\boldsymbol{\tau}, \boldsymbol{x} - \boldsymbol{x}'\rangle), \tag{3}$$

where the sum is over vectors of integers $\boldsymbol{\tau}$, and $\boldsymbol{x}, \boldsymbol{x}' \in [0,1)^d$ are the representation of elements of $\mathbb{T}^d$ as vectors of angles divided by $2\pi$, i.e., such that $\left(e^{2\pi i x_1}, \ldots, e^{2\pi i x_d}\right)$ and $\left(e^{2\pi i x'_1}, \ldots, e^{2\pi i x'_d}\right)$ are the corresponding elements of $\mathbb{T}^d$. Here, $C_\infty$ is the normalizing constant chosen so that $k(\boldsymbol{x}, \boldsymbol{x}) = \sigma^2$. The key difference with (1) is that the eigenpairs are now parameterized by vectors of integers, slightly simplifying the expression. A simple proof of this expression is given in Appendix B.

**Sphere.** The sphere manifold occurs in a number of settings: in particular, unit quaternions commonly used in robotics to represent the orientation of rigid bodies are a common example of the three-dimensional sphere $\mathbb{S}^3$. On the $d$-dimensional sphere $\mathbb{S}^d$ the squared exponential kernel is

$$k_{\infty,\kappa,\sigma^2}(\boldsymbol{x}, \boldsymbol{x}') = \frac{\sigma^2}{C_\infty} \sum_{n=0}^{\infty} c_{n,d}\, e^{-\frac{\kappa^2}{2}n(n+d-1)}\, \mathcal{C}_n^{(d-1)/2}\left(\cos(d_g(\boldsymbol{x}, \boldsymbol{x}'))\right), \tag{4}$$

where $C_\infty$ is a normalizing constant which ensures $k_{\infty,\kappa,\sigma^2}(\boldsymbol{x}, \boldsymbol{x}) = \sigma^2$; $c_{n,d}$ are explicit constants, $\mathcal{C}_n^{(\cdot)}$ are the Gegenbauer polynomials, and $d_g$ is the geodesic distance on the sphere [8]. This kernel is obtained by noting that the eigenfunctions of the spherical Laplace–Beltrami operator are the spherical harmonics, sums of which may be reformulated through the Gegenbauer polynomials—the resulting algebra, along with combining repeated eigenvalues, leads to the desired expression. This expression is a significant simplification compared to the general case: note in particular that the kernel factorizes and can be written as a function only of the geodesic distance $d_g$.

**Special orthogonal group.** The special orthogonal group $\mathrm{SO}(d)$ is a compact Riemannian manifold[2] of rotations of $d$-dimensional Euclidean space, most often represented as the set of orthogonal $d \times d$ matrices of determinant 1. $\mathrm{SO}(3)$ is widely used to represent orientations in the 3-dimensional physical space. This manifold may likewise be useful as a building block for representing other geometric objects. For instance, the product $\mathbb{R}^d \times \mathrm{SO}(d)$ may be used as a proxy to represent symmetric matrices via a non-unique decomposition $\mathbf{A} = \mathbf{U}\mathbf{D}\mathbf{U}^\top$, where $\mathbf{D} = \mathrm{diag}(\lambda_1, \ldots, \lambda_d)$, $\lambda_i \in \mathbb{R}$ and $\mathbf{U} \in \mathrm{SO}(d)$. A kernel on $\mathbb{R}^d \times \mathrm{SO}(d)$ may be designed as a product of kernels on $\mathbb{R}^d$ and on $\mathrm{SO}(d)$. By restricting the Euclidean part of the product kernel to $\mathbb{R}^d_{\geq 0}$ or $\mathbb{R}^d_{>0}$, we may obtain a kernel over positive (resp. semi)definite matrices, albeit not a canonical Matérn kernel.

By leveraging the group structure of $\mathrm{SO}(d)$ it is possible to express the corresponding heat kernel in terms of *group characters*. Specifically, we have that

$$k_{\infty,\kappa,\sigma^2}(\mathbf{X}, \mathbf{Y}) = \frac{\sigma^2}{C_\infty} \sum_{\pi} e^{-\frac{\kappa^2}{2}\lambda_\pi} d_\pi \chi_\pi(\mathbf{X}\mathbf{Y}^{-1}), \tag{5}$$

where the sum is over a certain set of tuples of nonnegative integers $\pi$ called the *highest weights* and $C_\infty$ is the normalizing constant ensuring that $k_{\infty,\kappa,\sigma^2}(\mathbf{X}, \mathbf{X}) = \sigma^2$: see Appendix B. Here, $\lambda_\pi$ are Laplacian eigenvalues, $\chi_\pi$ are certain functions called *characters* that depend only on eigenvalues of their argument, and $d_\pi = \chi_\pi(\mathbf{I})$. Partial sums of (5) corresponding to the most significant eigenvalues may be computed in closed form by means of the Weyl character formula and Freudenthal's formula for the eigenvalues of the Casimir operator. We provide technical details concerning this formula and its practical implementation in Appendix B and in the supplemented code respectively, while deferring a thorough discussion to future work.

---

[2]$\mathrm{SO}(d)$ is a Lie group, meaning that it is a group endowed with a structure of differentiable manifold: this manifold admits a canonical Riemannian structure induced by the Killing form.

**Hyperbolic space.** The hyperbolic space $\mathcal{H}^d$ is the unique simply-connected complete $d$-dimensional Riemannian manifold with a constant negative sectional curvature $-1$. An important property of this space is the exponential rate of growth of the volume of a ball with respect to its radius. Because of this, it is often used to hold embeddings of hierarchical data, such as trees or graphs [24, 37]. Although its potential to embed discrete data structures into a continuous space is well-known in the machine learning community, its application in robotics is presently scarce. In Section 4, we provide a toy example featuring a simple path planning on a 2D grid to illustrate how the hyperbolic manifold can be exploited in robotics.

Hyperbolic space is non-compact, thus the technique of Borovitskiy et al. [8] is unable to represent this kernel. Fortunately, explicit formulas for the corresponding heat kernel in arbitrary dimension $d$ are present in the literature. Take $\boldsymbol{x}, \boldsymbol{y} \in \mathcal{H}^d$ and denote by $\rho = \text{dist}_{\mathcal{H}^d}(\boldsymbol{x}, \boldsymbol{y})$ the geodesic distance between $\boldsymbol{x}$ and $\boldsymbol{y}$ in $\mathcal{H}(d)$. Then we have, as introduced in Grigoryan and Noguchi [19],

$$k_{\infty,\kappa,\sigma^2}^{\mathcal{H}(2)}(\boldsymbol{x}, \boldsymbol{y}) = \frac{\sigma^2}{C_\infty} \int_\rho^\infty \frac{s \exp(-s^2/(2\kappa^2))}{(\cosh(s) - \cosh(\rho))^{1/2}}\, \mathrm{d}s, \quad k_{\infty,\kappa,\sigma^2}^{\mathcal{H}(3)}(\boldsymbol{x}, \boldsymbol{y}) = \frac{\sigma^2}{C_\infty} \frac{\rho}{\sinh \rho} e^{-\frac{\rho^2}{2\kappa^2}} \quad (6)$$

for $d = 2$ and $d = 3$, and the following recurrence relation, known as Millson's formula, for $d > 3$:

$$k_{\infty,\kappa,\sigma^2}^{\mathcal{H}(d)}(\boldsymbol{x}, \boldsymbol{y}) = -\frac{\sigma^2}{C_\infty \sinh \rho} \frac{\partial}{\partial \rho} k_{\infty,\kappa,\sigma^2}^{\mathcal{H}(d-2)}(\boldsymbol{x}, \boldsymbol{y}). \quad (7)$$

**Manifold of positive definite matrices.** Positive definite matrices occur naturally in many physics related problems. In robotics, they are used to represent the stiffness and manipulability ellipsoids [3, 57] or as control gain matrices [33]. These matrices strongly affect the performance of robot motion and force controllers, and they are variables of interest for task-compatible trajectory optimization.

Symmetric positive definite (SPD) matrices may be endowed with different Riemannian structures. For instance, $\mathbf{A} \mapsto \exp(\mathbf{A})$ and $\mathbf{A} \mapsto \mathbf{A}\mathbf{A}^\top$ define one-to-one maps between symmetric matrices or lower triangular matrices with positive elements on the diagonal and SPD matrices, allowing one to transfer Euclidean structure and kernels to SPD matrices. Though simple and natural, these Riemannian structures are often too simplistic and poorly capture the geometry of SPD matrices [13].

Another approach is to consider the map $\mathbf{A} \mapsto \mathbf{A}\mathbf{A}^\top$ as a function from the space $\text{GL}(d)$ of invertible matrices to $\mathcal{S}_{++}^d$. This map is not one-to-one: specifically, $\mathbf{A}\mathbf{A}^\top = \mathbf{B}\mathbf{B}^\top$ if and only if $\mathbf{A} = \mathbf{B}\mathbf{Q}$ where $\mathbf{Q}$ is an orthogonal matrix: see Appendix B for details. To make this map one-to-one, we need to switch the domain to the quotient space of $\text{GL}(d)$ by the equivalence relation of being equal up to an orthogonal matrix. The resulting space $\text{GL}(d)/\text{O}(d)$ is a Riemannian manifold. The Riemannian structure defined this way is not Euclidean and the resulting manifold is non-compact, hence the method of Borovitskiy et al. [8] does not apply here either. Still, explicit formulas for the corresponding heat kernels can be found in literature for cases of $d = 2$ and $d = 3$, which are common dimensions in robotics problems.

Specifically, for $d = 2$ there is a relatively simple formula. Take $\mathbf{X}, \mathbf{Y} \in \mathcal{S}_{++}^2$, denote by $H_1 \geq H_2$ the singular values of the matrix $\mathbf{X}\mathbf{Y}^{-1}$, and $\alpha = H_1 - H_2$, then, by Sawyer [43], we have

$$k_{\infty,\kappa,\sigma^2}(\mathbf{X}, \mathbf{Y}) = \frac{\sigma^2}{C_\infty} \exp\left(-\frac{H_1^2 + H_2^2}{2\kappa^2}\right) \int_0^\infty \frac{(2s + \alpha) \exp(-s(s + \alpha)/\kappa^2)}{(\sinh(s) \sinh(s + \alpha))^{1/2}}\, \mathrm{d}s, \quad (8)$$

where the one-dimensional integral on the right-hand side may be evaluated numerically and $C_\infty$ is a normalizing constant ensuring that $k_{\infty,\kappa,\sigma^2}(\mathbf{X}, \mathbf{X}) = \sigma^2$. Additional details are given in Appendix B.

**Product kernels on manifolds.** If the Riemannian manifold $\mathcal{M}$ of interest is the product of $d$ manifolds $\mathcal{M} = \mathcal{M}_1 \times \ldots \times \mathcal{M}_d$, where each factor $\mathcal{M}_j$ is equipped with a Matérn kernel $k_{\nu,\kappa,\sigma^2}^{\mathcal{M}_j}$, then we can introduce the following product Matérn kernel as a kernel on the product manifold $\mathcal{M}$. This is done by writing

$$k_{\nu,\boldsymbol{\kappa},\sigma^2}^{\mathcal{M}}(\boldsymbol{x}, \boldsymbol{x}') = \sigma^2 k_{\nu,\kappa_1,1}^{\mathcal{M}_1}(x_1, x_1') \cdot \ldots \cdot k_{\nu,\kappa_d,1}^{\mathcal{M}_d}(x_d, x_d'), \quad (9)$$

where $\boldsymbol{\kappa} = (\kappa_1, \ldots, \kappa_d)$, $\boldsymbol{x} = (x_1, \ldots, x_d)$ and $\boldsymbol{x}' = (x_1', \ldots, x_d')$. Note that this enables automatic relevance determination in the product Riemannian setting. Because of this, it is an alternative and more expressive way to define a kernel on e.g. a torus or other product manifold. Note that the product of Matérn kernels does not generally coincide with the Matérn kernel on the product manifold, unless $\nu = \infty$ and $\kappa_1 = .. = \kappa_d$. The product of manifolds naturally arises when representing the pose of rigid bodies such as the end-effector of a robotic manipulator, which lies on $\mathbb{R}^3 \times \mathbb{S}^3$.

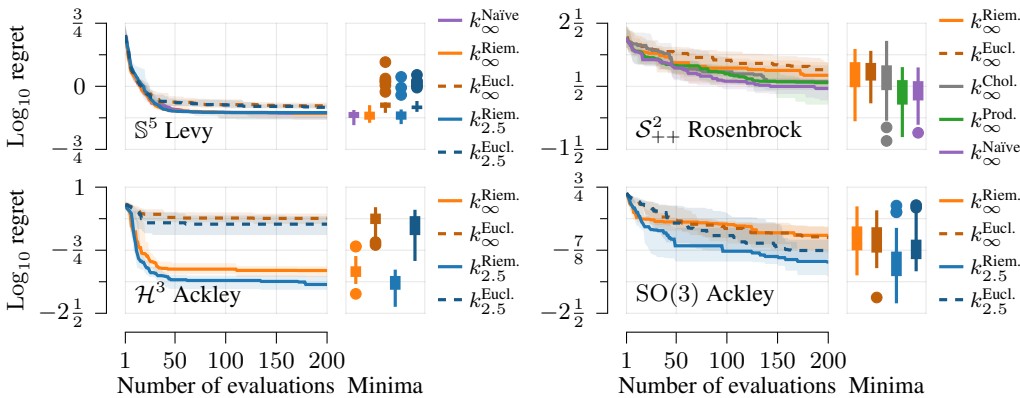

Figure 2: Logarithm of the regret for a set of benchmark test functions on $\mathbb{S}^5$, $\mathrm{SO}(3)$, $\mathcal{S}^2_{++}$, and $\mathcal{H}^3$.

## 4 Experiments

We evaluate the performance of geometry-aware Bayesian optimization via Riemannian Matérn kernels on a set of benchmark functions and robotics settings, using expected improvement [35] as the acquisition function. Each optimization process is repeated 30 times with 5 random initial samples. Our implementations employ GPyTorch [16], BoTorch [6] and Pymanopt [49] as well as the Robotics Toolbox [11]. Cost parameter values and additional experiments are given in Appendix A.

### 4.1 Benchmarks

We first study performance using benchmark test functions projected onto the manifolds $\mathbb{S}^5$, $\mathrm{SO}(3)$, $\mathcal{S}^2_{++}$, and $\mathcal{H}^3$. The search domain $\mathcal{X}$ is given by the full manifold, except for $\mathcal{S}^d_{++}$ where we restrict to matrices with eigenvalues $\lambda_i \in [0.001; 5]$. For each manifold, we compare geometry-aware Bayesian optimization against classical Bayesian optimization with Euclidean kernels using constrained optimization on the Euclidean space to minimize the acquisition function. For the sphere and SPD manifolds, we additionally compare against the scheme used by Jaquier et al. [27] featuring a naïve Riemannian squared exponential kernel, where Euclidean distance is replaced by the geodesic distance. For the SPD manifold, we consider two Riemannian kernels, namely (8), and a product of kernels $\mathbb{R}^2 \times \mathbb{S}^2$ which is applied on the eigendecomposition of SPD matrices. We also compare against an alternative implementation using the Cholesky decomposition of an SPD matrix $\mathbf{X} = \mathbf{L}\mathbf{L}^\mathsf{T}$, so that the resulting parameter is the vectorization of the lower triangular matrix $\mathbf{L}$.

Figure 2 shows performance results, including evolution of the median and the distribution of the logarithm of the simple regret of the final recommendation $\boldsymbol{x}_N$ after 200 iterations. We observe that geometry-aware algorithms generally match or outperform their Euclidean counterparts. These differences are more pronounced in higher dimensions, and for manifolds which more significantly depart from the Euclidean setting due to presence of curvature or other geometric features, with the hyperbolic manifold $\mathcal{H}^3$ showing the most pronounced difference. Surprisingly, the naïve Riemannian squared exponential kernel—which for certain length scales may become ill-defined— performs competitively in some settings: this may occur because small length scales keep it out of the problematic regime, and its simple form introduces less numerical error than series truncations or other approaches. Geometry-aware Matérn kernels ($\nu = 2.5$) outperform the squared exponential ones for the test functions with uneven landscapes, such as the Ackley function in Figure 2. Overall, we see that taking geometry into account generally results in faster, more data-efficient convergence and lower variance, at cost of more complex algorithmic implementation.

### 4.2 Robotics Experiments

Here, we evaluate the performance of geometry-aware Bayesian optimization on several simulated robotics scenarios. In the first experiment, we use Bayesian optimization as an orientation sampler aiming at satisfying the requirements defined by a cost function, similarly to the experiment presented in Jaquier et al. [27]. A velocity-controlled robot samples an orientation reference $\boldsymbol{x} = \hat{\boldsymbol{q}}$ around a

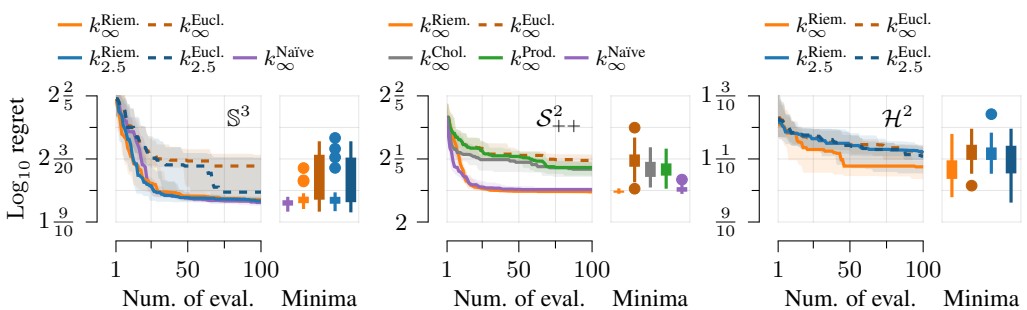

Figure 3: Logarithm of the regret for the orientation, task-compatible manipulability, and path planning problems.

prior orientation $\tilde{q}$. The objective of the robot is to minimize the absolute value of the difference between the prior and the current end-effector orientation $q$ with low joint torques $\tau$ and a high-volume manipulability ellipsoid, i.e., $f(q) = \sum w_q \| \mathrm{Log}_q(\tilde{q}) \|_1 + w_\tau \| \tau \|_2^2 - w_M \det(M)$, where $\det(M)$ is the determinant of the velocity manipulability ellipsoid, proportional to its volume.

Next, we consider a task-compatible manipulability optimization scenario, a common problem in robotics where most solutions do not exploit the geometry of manipulability ellipsoids [21, 29]. Here, an 8-degree-of-freedom planar robot is required to track a desired Cartesian velocity trajectory leading to a vertical line, while tracking a desired manipulability ellipsoid in its nullspace [26]. Optimization aims at finding the desired manipulability $x = \widehat{M} \in \mathcal{S}_{++}^2$ so that the end-effector trajectory jerk $\dddot{p}$ is minimized and the robot tracks the desired manipulabity aligning with the end-effector movement direction as proposed by Chiacchio [10]. This leads to $f(\widehat{M}) = \sum w_{\dddot{p}} \| \dddot{p} \|_2 + w_M \| \mathrm{Log}_M(\widehat{M}) \|_1 + w_t t^\top M t$, with $M$ the current manipulability and $t$ the unit vector tangential to the end-effector path.

Finally, we consider a simple path planning problem on $\mathbb{R}^2$ for a robot represented by a point. Our goal is to find a collision-free path between two points in the environment. To do so, we leverage geometry-aware Bayesian optimization to estimate paths for sampling, similarly to Marchant and Ramos [32], but introducing hyperbolic geometry. Specifically, we discretize the robot environment by defining a grid on $\mathbb{R}^2$ with 4-connected nodes, represented by a connected undirected graph $\mathcal{G} = (V, E)$, of $n$ vertices $V$ and edges $E$. A circle-shaped obstacle of unknown location occupies this environment: the robot finds this by sensing collisions while testing sampled paths. The graph $\mathcal{G}$ is mapped to $\mathcal{H}^2$ by learning embeddings in the Lorentz model of hyperbolic geometry, similarly to Nickel and Kiela [36]. At each iteration, the algorithm samples a set of $m$ points $\{ x_1, \ldots x_m \} \in \mathcal{H}^2$ representing a hyperbolic path of $m$ nodes, which are mapped back to a grid in $\mathbb{R}^2$ in order for the robot to navigate the environment using the resulting path. An optimal path corresponds to the shortest collision-free trajectory from a fixed starting location to the goal $x_g$. We minimize the cost $f(x_1, \ldots, x_m) = \sum w_g \, \mathrm{dist}^{x_g}(x_1, \ldots, x_m) + w_d \, \mathrm{dist}^t(x_1, \ldots, x_m)$, where $\mathrm{dist}^{x_g}(\cdot)$ is the distance to the goal $x_g$ (zero in no-collision cases), and $\mathrm{dist}^t(\cdot)$ is the total travel distance.

Figure 3 shows performance results. Here, we also see a moderate improvement in performance when using geometry-aware Bayesian optimization, mirroring the results on benchmark functions. Note that the gained performance improvements may depend on both the manifold geometry and cost function landscape. For instance, the additional orientation experiment reported in Appendix A show less-pronounced improvements as the low-dimensional nature may make the Euclidean and Riemannian kernels very similar. This contrasts with the results in the manipulability and path planning problems, whose geometries depart more from the Euclidean setting.

## 5 Conclusion

We study geometry-aware Bayesian optimization for robotics, and propose techniques for computing geometry-aware kernels on various Riemannian manifolds occurring in robotics applications. This is achieved by studying how general expressions for Matérn kernels based on stochastic partial differential equations simplify for the manifolds of interest. This provide practitioners with explicit formulas, enabling the use of geometry-aware Bayesian optimization in novel settings. We demonstrate our techniques both on benchmarks and robotics examples to showcase improved performances and potential use cases. We hope these contributions enable greater use of geometry among roboticists.

**Acknowledgments**

NJ and TA were supported by the Carl Zeiss Foundation under the project JuBot (Jung Bleiben mit Robotern). VB was supported by the Ministry of Science and Higher Education of the Russian Federation, agreement № 075-15-2019-1620 and by "Native towns", a social investment program of PJSC Gazprom Neft.

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
