# OpenReview forum: "Geometry-aware Bayesian Optimization in Robotics using Riemannian Matérn Kernels"
_robot-learning.org/CoRL/2021/Conference — CoRL2021 Poster_

### Official Review · Reviewer_f3EN · 2021-07-23

**Originality:** Fair
**Technical Quality:** Good
**Clarity Of Presentation:** Very Good
**Impact:** 2

**Recommendation:**

Weak Accept: I recommend accepting the paper, but will not argue for my recommendation if the majority of other reviewers have a different opinion.

**Summary:**

This paper studies the application of Bayesian optimization with Riemannian Matern kernels in various robotics applications where the search domain for BO lies in a Riemannian manifold instead of the commonly assumed Euclidean space.

**Issues:**

As mentioned above, my most major concern is regrading the technical contribution. In particular, it's unclear to me which part of the paper can be considered as novel contributions from this paper compared with previous works (especially the most related works of [17] and [7]).


**Reviewer Expertise:**

Good: General knowledge of the area

**Strengths And Weaknesses:**

Strengths:
- The paper covers a large variety of robotics applications with various Riemannian manifolds, and hence can potentially contribute useful ideas to a large number of robotics tasks.
- The paper is clearly motivated and in general well written.

Weaknesses:
- My most major concern is regarding the significance of the contribution of the current paper. In particular, this paper feels more like a case study on the use of existing Riemannian Matern kernels in different robotics applications. So I wonder what are the novel contributions of this paper, especially when compared with the most related works of [17] and [7]?
- line 170: if I understand correctly, the expressions of all the kernels introduced in the remainder of this section are based on the approach of heat kernels (equation 3)? So the method of equation (2) is not used at all? Please clarify.
- Experiments, lines 269-271: which method in the figures correspond to this baseline? "Approx. Riem. SE"?
- Figure 2: for the SO(3) and $\mathcal{H}^3$ spaces, why only two methods are compared here?
- Figure 2 and 3: It seems that the "Riem. SE" method works the best in general across all figure. Does this suggest maybe there is no need for Matern kernels in these applications?
- Figure 2: I think the figure in its current form is very hard to read. Since you compare with different baselines in different sub-figures, I would suggest having a separate legend for every sub-figure

Minor issues:
- line 71: symmetric-positive-definite: I think it's usually semi-definite instead of definite

**Summary Of Recommendation:**

This paper provides a comprehensive study of the use of Bayesian optimization with Riemannian kernels in a variety of robotics tasks. My most major concern is regarding the technical novelty of the paper. Please clarify this, as well as my other concerns listed in the section "Strengths and Weaknesses" above.

---

> ### Author Response · Authors · 2021-08-26
> **Answer to the comments of Reviewer f3EN**
>
> Thank you very much for your review! We are delighted to hear our work was described as "clearly-motivated" and "well-written"!
> Below, we address the key issues raised in the review.
>
> > As mentioned above, my most major concern is regrading the technical contribution. In particular, it's unclear to me which part of the paper can be considered as novel contributions from this paper compared with previous works (especially the most related works of [17] and [7]).
>
> Thank you for raising this important point. This is addressed in detail in our response to the meta-reviewer, which provides a complete statement of the contributions and differences between our work and Borovitskiy et al. [7] and Jaquier et al. [17].
>
> In short, the kernel formulation is indeed one of the key contributions: as opposed to Jaquier et al. [17], the proposed kernel formulations are *(a)* theoretically sound and valid for all length scales, and *(b)* extend to the more general Matérn kernels on Riemannian manifolds. On the other hand, in contrast to Borovitskiy et al. [7] who considered only compact manifolds (and whose technique does not apply in the non-compact setting), we *(a)* extend the formulation to non-compact manifolds (symmetric positive-definite (SPD) matrices, and hyperbolic spaces) in a technically sound manner using a connection between Matérn and heat kernels, *(b)* provide more explicit formulas for the special orthogonal group, and *(c)* illustrate their use in robotics applications.
>
> > line 170: if I understand correctly, the expressions of all the kernels introduced in the remainder of this section are based on the approach of heat kernels (equation 3)? So the method of equation (2) is not used at all? Please clarify.
>
> The two expressions presented at the beginning of Section 3 allows us to connect squared exponential kernels with Matérn kernels by viewing the latter as certain integrals of the former. Once the squared exponential kernel is known for a given manifold, this connection is used in order to *define* the Matérn formulation for this manifold. Therefore, we focus on squared exponential exponential kernels in the remaining of Section 3, as Matérn kernels may then be directly obtained by using the aforementioned formulas. These were used to obtain, e.g., the Matérn kernel for the hyperbolic space in our experiments, which is not covered by previous theory. We precised this point in our paper (Sections 2.1 and 3).
>
> > Experiments, lines 269-271: which method in the figures correspond to this baseline? "Approx. Riem. SE"?
>
> The "Approx. Riem. SE" corresponds to the naïve generalization of the SE kernel to Riemannian manifolds presented in Jaquier et al. [17], which is not a positive semi-definite kernel for all length scales and requires length scale restrictions which are determined empirically.
>
> > Figure 2: for the SO(3) and spaces, why only two methods are compared here?
> >
> > Figure 2 and 3: It seems that the "Riem. SE" method works the best in general across all figure. Does this suggest maybe there is no need for Matern kernels in these applications?
> >
> > Figure 2: I think the figure in its current form is very hard to read. Since you compare with different baselines in different sub-figures, I would suggest having a separate legend for every sub-figure
>
> **Added comparisons**. We have added comparisons with the Riemannian Matérn kernel and the Euclidean Matérn kernels, which were indeed missing. We thank the reviewer for pointing out this omission.
>
> **Advantages of Matérn kernels**. Our response to the meta-reviewer includes a complete discussion on cases where there is an advantage to be gained by using Riemannian Matérn kernels over squared exponential kernels. These are experimentally illustrated in the updated Figure 2: the Riemannian Matérn kernel ($\nu=2.5$) outperforms the Riemannian squared exponential kernel for the Ackley benchmark function on $\text{SO}(3)$ and $\mathcal{H}^3$.
>
> **Legends**. We thank the reviewer for the useful suggestions on the presentation of our figures. We changed their format and added one legend per sub-figure in all our experiments.
>
> > Minor issues:
> > line 71: symmetric-positive-definite: I think it's usually semi-definite instead of definite
>
> In our applications, we consider the manifold of symmetric positive-definite matrices, which are matrices with strictly positive eigenvalues. Other applications may include null eigenvalues: in that case, the manifold of positive semi-definite matrices should be used. Kernels for those applications can be obtained using similar techniques to the ones presented.

---

### Official Review · Reviewer_znr9 · 2021-07-24

**Originality:** Very Good
**Technical Quality:** Very Good
**Clarity Of Presentation:** Excellent
**Impact:** 4

**Recommendation:**

Strong Accept: I recommend accepting the paper and will argue for my recommendation even if other reviewers hold a different opinion.

**Summary:**

This paper focuses on the problem of designing geometry-based kernels for Gaussian Process optimization. The kernels are derived for several well-known manifolds that appears naturally in several robotic applications. The geometry-based kernels are compared against the Euclidean-based kernels to validate the performance of the proposed kernels.

**Issues:**

N/A.

**Reviewer Expertise:**

Fair: Some knowledge of the area

**Strengths And Weaknesses:**

Strengths:
The paper is well written paper, in general, with solid ideas and contributions. In particular, the proposed kernels are sound and easy to compute with detailed analysis provided in the appendix. The numerical results provide a clear evidence of the gained performance when geometry-based kernels are used.


**Summary Of Recommendation:**

Solid paper with clear application to robotic applications.

---

> ### Author Response · Authors · 2021-08-26
> **Answer to the comments of Reviewer znr9**
>
> Thank you very much for your time reviewing our work! It was exactly our goal to provide kernels which are "sound and easy to compute" so that practitioners can consider geometry-aware Bayesian optimization for their use case. We appreciate the positive feedback about our paper!
>
> If any particular issues in need of further clarification arise that were not apparent at the time of the review, we are happy to respond in follow-up comments as needed.

---

### Official Review · Reviewer_3LxP · 2021-07-24

**Originality:** Good
**Technical Quality:** Good
**Clarity Of Presentation:** Good
**Impact:** 3

**Recommendation:**

Weak Accept: I recommend accepting the paper, but will not argue for my recommendation if the majority of other reviewers have a different opinion.

**Summary:**

This paper is concerned with endowing GP models the ability to consider the geometry of the search space (e.g. the domain of the function being optimised lies on a manifold). Developing these geometry aware GP models is non-trivial, thus the authors leverage existing tools for formulating valid kernels that respect various problem geometries. The main contribution is extending the work of a previous paper [17] by applying these geometry-aware GP models to additional robotics problems and formulating their appropriate kernels.

**Issues:**

- The robotic experimental results are not as convincing, for example some of the euclidean kernels seem to perform very similarly.
- It would be useful to show computation time to assess whether these algorithms could be run in real-time.

**Reviewer Expertise:**

Good: General knowledge of the area

**Strengths And Weaknesses:**

- The robotic experimental results are not as convincing, for example some of the euclidean kernels seem to perform very similarly.
- It would be useful to show computation time to assess whether these algorithms could be run in real-time.

**Summary Of Recommendation:**

This paper provides an incremental contribution to [17] by extending its kernel formulations to additional manifold types relevant to various robotics problems. In particular, these formulations are tailored to their Riemannian structure and overcome the issue of having to choose a small enough length scale; for example, required when using the more general geodesic SE kernel. Although the paper does not present any new fundamental ideas, the derivations are interesting and non-trivial and could be useful for practitioners working on similar problems.

---

> ### Author Response · Authors · 2021-08-26
> **Answer to the comments of Reviewer 3LxP**
>
> Thank you very much for your review! We are delighted to hear that our derivations are "interesting and non-trivial"---in fact, studying the machinery for handling non-compact manifolds and cases of interest in robotics like the special orthogonal group is a key part of our contribution. Please see our response to the meta-reviewer for more details about the ideas we introduce provide additional contribution on top of prior works.
>
> > The robotic experimental results are not as convincing, for example some of the euclidean kernels seem to perform very similarly. It would be useful to show computation time to assess whether these algorithms could be run in real-time.
>
> We agree that the difference between geometry-aware and Euclidean Bayesian optimization may be limited for some experiments, which are sufficiently close to Euclidean that geometry plays a more minor role. This will occur in low-dimensional problems with simple geometries and featuring relatively simple cost functions. Thus the difference are least pronounced for the orientation experiment of Section 5.2, where the low-dimensional nature may make the Euclidean and Riemannian kernels very similar. We believe it is important to have an example like this, both as a sanity check that our method still works in classical settings, as well as to understand when geometry-aware methods are beneficial.
>
> **New orientation experiment**. We have added a second orientation experiment featuring a more complex cost function in Appendix A.3, for which the Riemannian BO algorithm clearly outperform its Euclidean counterparts.
>
> **Clarification of other experiments**. Compared to the orientation experiment, geometric differences are more pronounced in the manipulability and path planning problems, which depart more from the Euclidean setting (see Fig. 3 middle and right, as well as the additional manipulability experiment added in Appendix A.3).
> They are also more pronounced in higher dimensions. We clarify this point in Section 4 and added additional robotics experiment in Appendix A.3 to further illustrate it.
>
> **Added comparisons**. We have added additional comparisons for the SPD manifold in Figures 2 and 3, as well as in Figures 5 and 6 in Appendix A.
>
> > It would be useful to show computation time to assess whether these algorithms could be run in real-time.
>
> The computational costs of geometry-aware Bayesian optimization are similar to those of Euclidean Bayesian optimization.
> These are heavily dependent on the details of how the Gaussian process model used is trained, including the implementation.
>
> The main difference is that in geometric settings, pointwise evaluation of kernels on computation of infinite sums or low-dimensional integrals, depending on the particular manifold (and, in particular, whether or not it is compact).
> While these computations can be more involved than those of the Euclidean case, they are also highly parallelizable, and therefore amenable to acceleration on GPUs.
>
> We believe that the potential for running geometry-aware Bayesian optimization in real-time is similar to that of Euclidean Bayesian optimization.
> Creation of software libraries that make this practical, for both the geometric and non-geometric setting, is a promising direction for future work.

---

### Official Review · Reviewer_6VZw · 2021-07-25

**Originality:** Good
**Technical Quality:** Good
**Clarity Of Presentation:** Good
**Impact:** 3

**Recommendation:**

Weak Accept: I recommend accepting the paper, but will not argue for my recommendation if the majority of other reviewers have a different opinion.

**Summary:**

This paper presents a Bayesian optimisation framework for optimisation problems over Riemannian manifolds commonly appearing in robotics problems. This work extends the Geometry-aware Bayesian optimisation (GABO) framework [17] by including techniques to derive Matérn kernels for manifolds where a squared-exponential kernel is defined. The paper includes expressions for the sphere, SO(d), spaces of positive-definite matrices and hyperbolic spaces, which are common in robotics planning problems. Experimental results demonstrate the performance of the proposed methodology on synthetic functions and simulated robotics problems.

**Issues:**

1. The contrast with Jaquier et al. [17]'s original GABO is not clear at first. Since the proposed framework is mostly similar, I believe a more clear statement of contributions and differences with respect to the original framework would help assessing the novelty in this paper. Since Jaquier et al. only used the squared exponential kernel, I wonder if the main difference lies in the new kernels formulation?

2. Performance differences w.r.t. standard Euclidean kernels in robotics experiments (Figure 3) do not seem very evident, except for the planning problem, which only had one baseline and is apparently in a very simplified setting.

3. Experimental comparisons against the original GABO are restricted to the orientation sampler (sphere manifold), though that framework had also been tested at least on the positive-definite matrices manifold [17].


**Reviewer Expertise:**

Very good: Comprehensive knowledge of the area

**Strengths And Weaknesses:**

The paper is mostly well written and clear in its presentation structure. I believe the expressions provided for Matérn on common robotics manifolds can be very useful to the community, especially to those working with kernel methods and planning over manifolds. However, the paper comes with a few issues which make it hard to assess the significance of its contribution.

**Summary Of Recommendation:**

My main concern with this paper is that it seems incremental when compared to the original GABO framework [17] and the experimental results do not seem to show much improvement.

---

> ### Author Response · Authors · 2021-08-26
> **Answer to the comments of Reviewer 6VZw**
>
> Thank you for taking the time to review our work! We are delighted to hear that our work was "well-written", "clear in its presentation structure", and "can be very useful to the community"! Below, we address some of the key concerns raised as part of the review.
>
> > The contrast with Jaquier et al. [17]'s original GABO is not clear at first. Since the proposed framework is mostly similar, I believe a more clear statement of contributions and differences with respect to the original framework would help assessing the novelty in this paper. Since Jaquier et al. only used the squared exponential kernel, I wonder if the main difference lies in the new kernels formulation?
>
> Thank you for raising this important point. This is addressed in detail in our response to the meta-reviewer, which provides a complete statement of the contributions and differences between our work and Borovitskiy et al. [7] and Jaquier et al. [17].
>
> In short, the kernel formulation is indeed one of the key contributions: As opposed to Jaquier et al. [17], the proposed kernel formulations are *(a)* theoretically sound and valid for all length scales, and *(b)* extend to the more general Matérn kernels on Riemannian manifolds. On the other hand, in contrast to Borovitskiy et al. [7] who considered only compact manifolds (and whose technique does not apply in the non-compact setting), we *(a)* extend the formulation to non-compact manifolds (e.g. symmetric positive-definite (SPD) matrices, and hyperbolic spaces) in a technically-sound manner using a connection between Matérn and heat kernels, *(b)* provide more explicit formulas for the special orthogonal group, and *(c)* illustrate their use in diverse robotics applications.
>
> > Performance differences w.r.t. standard Euclidean kernels in robotics experiments (Figure 3) do not seem very evident, except for the planning problem, which only had one baseline and is apparently in a very simplified setting.
>
> We agree that the difference between geometry-aware and Euclidean Bayesian optimization may be limited for some experiments, which are sufficiently close to Euclidean that geometry plays a more minor role. This will occur in low-dimensional problems with simple geometries and featuring relatively simple cost functions. Thus the difference are least pronounced for the orientation experiment of Section 5.2, where the low-dimensional nature may make the Euclidean and Riemannian kernels very similar. We believe it is important to have an example like this, both as a sanity check that our method still works in classical settings, as well as to understand when geometry-aware methods are beneficial.
>
> **New orientation experiment**. We have added a second orientation experiment featuring a more complex cost function in Appendix A.3, for which the Riemannian BO algorithms clearly outperform its Euclidean counterparts.
>
> **Clarification of other experiments**. Compared to the orientation experiment, geometric differences are more pronounced in the manipulability and path planning problems, which depart more from the Euclidean setting (see Fig. 3 middle and right, as well as the additional manipulability experiment added in Appendix A.3).
> They are also more pronounced in higher dimensions. We clarify this point in Section 4 and added additional robotics experiment in Appendix A.3 to further illustrate it.
>
> > Experimental comparisons against the original GABO are restricted to the orientation sampler (sphere manifold), though that framework had also been tested at least on the positive-definite matrices manifold [17].
>
> **Added comparisons**. We have added additional comparisons for the SPD manifold in Figures 2 and 3, as well as in Figures 5 and 6 in Appendix A.

---

### Meta-Review · Area_Chair_HSe3 · 2021-08-06

**Recommendation:** Accept (Poster)
**Confidence:** 5

**Metareview:**

The paper describes a Bayesian optimisation methodology on manifolds with Matern Kernels. The main contribution is the derivation of BO for Riemannian Matern kernels with interesting applications in robotics.

As pointed out by reviewers 6VZw, 3LxP and f3EN, there are two issues with the paper the authors should try to address in the rebuttal:

1. The novelty in the paper is not clear given [17] and the derivation of the GPs on Riemannian Manifolds with Matern kernels in [7]. What are the specifics of the contribution that goes beyond applying ideas in [7] into [17]?

2. The experiments do not demonstrate a significant advantage of Matern kernels, despite their generality, compared to squared exponential on manifolds. Given that the SE kernel is a special case within the Matern family, would you be able to provide examples of problems where the cost (implicit) function requires a less smooth kernel to be modelled properly?

========== Post discussion update ============

I thank you the authors for addressing the reviewers concerns and in improving the paper with a better explanation of the novelty in this work. The reviewers now agree that this paper can be accepted. This is also my recommendation.

---

> ### Author Response · Authors · 2021-08-26
> **Answer to the meta review and general answer to all reviewers - Part 1**
>
> We would like to thank the reviewers and the area chair for their highly useful
> recommendations on the previous version of the manuscript. We have addressed the reviewers comments and improved our paper accordingly, with the principal changes highlighted in blue.
>
> We first generally address the comment shared by all the reviewers concerning *(1)* the novelty in the paper, and *(2)* the advantages of Matérn kernels compared to square exponential kernels on manifolds. Then, we address specific comments individually.
>
> > The novelty in the paper is not clear given [17] and the derivation of the GPs on Riemannian Manifolds with Matern kernels in [7]. What are the specifics of the contribution that goes beyond applying ideas in [7] into [17]?
>
> (1) **Novelty**.
>
> **Contributions made solely by this work**. Our work contributes two key ideas not present in the prior works: *(1)* kernels for manifolds not previously considered, such as *(a)* the special orthogonal group and *(b)* non-compact manifolds including positive definite matrices and hyperbolic spaces, and *(2)* novel uses of geometry-aware Bayesian optimization for robotics, including optimization of manipulability ellipsoids and hyperbolic path planning.
>
> Since *(1a)* and *(1b)* involve development of new theory not covered by prior work, and *(2)* are of direct interest, we therefore believe that the present work goes beyond the combination of the two aforementioned papers.
>
> **Detailed comparison to Jaquier et al. [17]**. This paper focuses primarily on Bayesian optimization. The kernels proposed by Jaquier et al. [17] are a *naïve* generalization of the Euclidean squared exponential kernels for the sphere and SPD manifolds. By *naïve*, we mean they are not theoretically guaranteed positive-definite, and thus valid length scale parameters need to be determined experimentally.
>
> Our more general formulation avoids the need to determine valid length scales as in Jaquier et al. [17], provides theoretical guarantees of positive-definiteness, and provides not only squared exponential, but Matérn kernels for which smoothness optimization is possible.
> We also consider additional manifolds, such as hyperbolic space.
>
> **Detailed comparison to Borovitskiy et al. [7]**. This paper focuses entirely on geometric Gaussian processes. Borovitskiy et al. [7] provide theoretically-sound Matérn kernels on *compact* manifolds, i.e., on the sphere and torus, and the developed theory relies very heavily on compactness (otherwise, the main results involving sums infinite series are not true).
>
> We build on this work, but we also study non-compact manifolds which arise in robotics, such as the manifold of positive-definite matrices and hyperbolic space. For these non-compact manifolds, we define Matérn kernels by using a connection between Riemannian squared exponential kernels and heat kernels in the sense of fundamental solutions of heat equations.
>
> In total, we obtain *(a)* Riemanian Matérn kernels for non-compact manifolds, including hyperbolic space and manifold of symmetric positive definite matrices, but also *(b)* useful ways of expressing kernels on certain compact manifolds such as the special orthogonal group, which are of interest in their own right.
> Building such kernels enables us to study their use in a wider set of robotic applications.
>
> **Concluding remarks.** We would like to emphasize the fact that the proposed kernels allows us to bring a novel view on some robotics problems. For examples, we introduce novel perspectives on path search problems by using the hyperbolic manifold, and on manipulability optimization by considering the manifold of positive-definite matrices.
>
> **Changes.** To make these contributions more clear to the readers, we reformulated the end of the introduction in order to clarify the contributions of the proposed approach. We also emphasized these contributions in Sections 2.1 and 3.

---

> > ### Author Response · Authors · 2021-08-26
> > **Answer to the meta review and general answer to all reviewers - Part 2**
> >
> > > The experiments do not demonstrate a significant advantage of Matern kernels, despite their generality, compared to squared exponential on manifolds. Given that the SE kernel is a special case within the Matern family, would you be able to provide examples of problems where the cost (implicit) function requires a less smooth kernel to be modelled properly?
> >
> > (2) **Advantages of Matérn kernels.**
> >
> > **Practical Benefit.** Since the most effective choice of kernel will depend on the details of the application, we do not believe and do not seek to demonstrate that the Matérn kernel is superior to the squared exponential kernel.
> > Instead, we would like to provide practitioners with the ability to consider both and decide what works better for them: one of our goals is providing the tools to make this practical.
> > We believe significant value is added by having the Matérn kernel as an option, given its easy-to-understand hyperparameters and wide use in areas such as spatial statistics.
> >
> > **Comparison.** As in the Euclidean case, the performance of the squared exponential and Matérn kernel depend heavily on the smoothness of the function to be modeled by the Gaussian process.
> >
> > Squared exponential kernels result in infinitely-differentiable processes, and are thus well adapted to model smooth functions. In contrast, Matérn kernels are better adapted for uneven functions (lower smoothness parameters $\nu$ result in less-differentiable processes). In practice, choosing between a squared exponential and a Matérn kernel for a given task depends heavily on the user's a priori knowledge on the given task or, alternatively, on cross-validation.
> >
> > As an example, the Riemannian Matérn kernel ($\nu=2.5$) outperforms the Riemannian squared exponential kernel on the Ackley benchmark function on $\text{SO}(3)$ and $\mathcal{H}^3$ (theses comparisons were added in Fig. 2).
> >
> > In total, we believe that the ability to control prior smoothness adds a valuable tool to the practitioner's toolbox, compared to not having this option.

---

### Decision · Program_Chairs · 2021-09-13

**Decision:**

Accept (Poster)

**Comment:**

The paper describes a Bayesian optimisation methodology on manifolds with Matern Kernels. The main contribution is the derivation of BO for Riemannian Matern kernels with interesting applications in robotics.

As pointed out by reviewers 6VZw, 3LxP and f3EN, there are two issues with the paper the authors should try to address in the rebuttal:

1. The novelty in the paper is not clear given [17] and the derivation of the GPs on Riemannian Manifolds with Matern kernels in [7]. What are the specifics of the contribution that goes beyond applying ideas in [7] into [17]?

2. The experiments do not demonstrate a significant advantage of Matern kernels, despite their generality, compared to squared exponential on manifolds. Given that the SE kernel is a special case within the Matern family, would you be able to provide examples of problems where the cost (implicit) function requires a less smooth kernel to be modelled properly?

========== Post discussion update ============

I thank you the authors for addressing the reviewers concerns and in improving the paper with a better explanation of the novelty in this work. The reviewers now agree that this paper can be accepted. This is also my recommendation.